# The negative emission potential of alkaline materials

Phil Renforth[1]

7 billion tonnes of alkaline materials are produced globally each year as a product or by-product of industrial activity. The aqueous dissolution of these materials creates high pH solutions that dissolves $CO_2$ to store carbon in the form of solid carbonate minerals or dissolved bicarbonate ions. Here we show that these materials have a carbon dioxide storage potential of 2.9–8.5 billion tonnes per year by 2100, and may contribute a substantial proportion of the negative emissions required to limit global temperature change to <2 °C.

---

[1] School of Engineering and Physical Sciences, Heriot-Watt University, Edinburgh EH14 4AS, UK. Correspondence and requests for materials should be addressed to P.R. (email: P.Renforth@hw.ac.uk)

I n addition to substantial cuts in greenhouse gas emissions, humanity may need to remove a large amount of carbon dioxide from the atmosphere to avoid climate change. The ability to remove multiple Gt of $CO_2$ every year is an important feature of integrated assessment models and particularly those that result in global mean surface temperature increases less than 2 °C[1–3]. By 2100, this cumulative negative emission requirement may be on the order of 100 to 1000 $GtCO_2$ (~1 to 15 $GtCO_2\,yr^{-1}$) in 1.5 °C pathways with little or no overshoot and is mostly met by biomass energy carbon capture and storage and afforestation[3]. There is uncertainty in the potential of most negative emission technologies, which may constrain the rate and extent of their scale-up[4,5]. Technologies that propose to remove $CO_2$ from the atmosphere by chemical reaction with natural or artificial minerals are included in literature assessments of negative emissions, but have received substantially less attention compared to other proposals[6].

Here we consider the potential of negative emissions within existing global industries. Particularly by weathering materials produced from the manufacturing of steel, aluminium, cement, lime, nickel, and from the combustion of coal or biomass. The alkaline materials produced from these activities include blast furnace and steel slag, red mud, cement kiln dust, concrete in building products and demolition waste, ultramafic waste rock and mine tailings and fuel ashes/residue. These materials contain silicate and hydroxide minerals that can dissolve in water and react with $CO_2$ to produce aqueous bicarbonate ions. If these bicarbonate ions were conveyed to the ocean (e.g., in river water), they would contribute to ocean alkalinity, potentially ameliorating some of the impacts of ocean acidification, and remain in solution for >100,000 years[7]. This enhanced weathering process[7,8] (Eqs. (1) and (2)) requires that the bicarbonate ions are stored in the ocean, otherwise additional mineral dissolution would lead to the formation of solid carbonate minerals in which some of the $CO_2$ may be trapped for millions of years (known as mineral carbonation, e.g.[9], Eqs. (1 + 3) and Eqs. (2 + 3)).

$$Ca(OH)_2 + 2CO_2 \rightarrow Ca^{2+} + 2HCO_3^-. \tag{1}$$

$$CaSiO_3 + 2CO_2 + H_2O \rightarrow Ca^{2+} + 2HCO_3^- + SiO_2. \tag{2}$$

$$Ca^{2+} + 2HCO_3^- \rightarrow CaCO_3 + CO_2 + H_2O. \tag{3}$$

While both mechanisms result in carbon dioxide sequestration, almost twice as much $CO_2$ is removed through enhanced weathering compared to mineral carbonation (the ratio is closer 1.5–1.8[7]), which is highly desirable when material supply is limited. However, there is little research that examines the environmental consequences of increasing ocean alkalinity, and particularly the impact of harmful trace elements that are present in some alkaline materials[7]. While the residence time of bicarbonate ions in the ocean is effectively permanent, this may be reduced if alkalinity is elevated[7]. As such, storage of carbon dioxide as a mineral carbonate may be the preferred mechanism, which would also reduce the potential for environmental harm[10,11]. Both mechanisms have been included in this assessment of storage potential.

Carbon dioxide sequestration has been demonstrated using these materials in elevated temperature and high $CO_2$ pressure (HTP) reactor experiments[12,13]. However, there is also evidence that atmospheric $CO_2$ is sequestered under ambient conditions[14–16]. These materials are created by emission intensive industries, and it is therefore reasonable to suggest that the carbon sequestration potential of the by-products should be used to offset some of these emissions. For instance, the steel industry creates approximately 2200 $kg\,CO_2\,t^{-1}$ of steel, which equates to around 12,000 $kg\,CO_2\,t^{-1}$ of by-product slag

(Table 1, column a and Supplementary Notes 1, 2, and 3). The Intergovernmental Panel on Climate Change (IPPC)[3] considers that extensive mitigation (e.g., decarbonised energy, carbon capture and storage, energy efficiency improvements) may be able to reduce the emissions intensity to 200–500 $kg\,CO_2\,t^{-1}$ of steel (or ~1000 $kg\,CO_2\,t^{-1}$ slag, column b). Some postulate that the integration of hydrogen into steel making may reduce emissions to <60 $kg\,CO_2\,t^{-1}$ (<300 $kg\,CO_2\,t^{-1}$ slag)[17]. The carbon dioxide capture potential through mineral carbonation or enhanced weathering of slag is 368–620 $kg\,CO_2\,t^{-1}$. Therefore, only a small proportion of current emissions from most of these industries can be offset by the carbon sequestration in alkaline wastes/by-products. However, by pursuing extensive mitigation together with atmospheric carbon dioxide sequestration in alkaline materials, it may be possible to create industries with net negative emissions, and thus contribute to limiting temperature change to <2 °C.

Here we examine the potential of alkaline materials to remove $CO_2$ from the atmosphere by forecasting production to 2100, and show that a large proportion of the future negative emission requirements may be met through weathering or carbonation of these materials.

## Results

**The potential of alkaline material streams.** Manufacturing iron and steel produces a range of alkaline wastes/by-products that are rich in oxide, hydroxide and silicate minerals and glasses, collectively referred to as slag. The physical and chemical properties, and the environmental behaviour, of slag depends on the raw materials, the process of creating iron and steel and the method of disposal. Blast furnace slag is commonly used as secondary aggregate, pozzolan or agricultural lime[18–20]. However, due to the higher concentrations of oxides and hydroxides, slags from steel production are typically stockpiled[21]. These sites have highly alkaline leachates with pH > 10[21] and can pose environmental issues via extreme pH and potential metal pollution[22]. $CO_2$ uptake buffers the waters back towards circum-neutral pH, which also limits metal solubility. HTP mineral carbonation experiments have shown 50–75% conversion of slag over 30 min[12]. However, studies investigating legacy deposits have demonstrated $CO_2$ uptake and carbonate precipitation within the drainage waters and surrounding environments[21].

Cement is produced by heating limestone ($CaCO_3$) in a kiln with a source of silicon (clay/shale) to produce metastable calcium silicate minerals (clinker, e.g., $Ca_2SiO_4$). The clinker is hydrated during construction to produce mortar and concrete. These materials, together with by-product cement kiln dust, have been successfully carbonated in HTP experiments[11,13], during curing under elevated $CO_2$ concentrations[23], during the life of the structure[24], when mixed into urban soils following demolition[14], or within leachate management systems of landfill[25].

Like cement, lime is produced by heating limestone in a kiln but is subsequently used in numerous industries (Supplementary Note 4 and Supplementary Table 1[26]). Of the lime produced in the United States and European Union, 30–40% is used by the steel industry as a fluxing agent, 14% is used in other industries (sugar refining, glass, paper, precipitated calcium carbonate), 10–20% is used in construction and 16–24% is used for environmental remediation/treatment (flue gas desulphurisation, water treatment, acid mine drainage). Approximately 20% of lime is used in activities that exploit reactions with $CO_2$ (e.g., regenerating NaOH in the Kraft process) or weathering (e.g., agricultural liming). Approximately 14% of the lime is used in activities that do not have an explicit reaction with $CO_2$, but it may be possible to engineer this within the life cycle of the material (e.g., soda lime glass, soil stabilisation).

**Table 1 Carbon production intensities and sequestration potential of highly alkaline materials, by-products and wastes**

| Material | 2010 $CO_2$ intensity[a] | 2050 $CO_2$ intensity[b] | Carbonation potential[c] | Measured carbonation[d] | Enhanced weathering potential[e] | Carbon offset recycling/reuse[f] |
|---|---|---|---|---|---|---|
| Blast furnace slag | 12,000 | 2700–4300 (286–1080)[i] | 413 ± 13 | 90–230 | 620 ± 19 | ~100. Up to 700 in high substitution specialised cements. <5 as aggregate |
| Basic oxygen furnace slag | | | 402 ± 17 | 50–540 | 602 ± 25 | |
| Electric arc furnace slag | | | 368 ± 10 | | 552 ± 15 | |
| Ordinary portland cement | 800 | 200–400 (100–200)[i] | 510 | 300 | 773 | — |
| Cement kiln dust | 6900[g] | 1700–3500 | 330 ± 12 | 82–260 | 530 ± 21 | ~0 Recycled into kiln |
| Construction and demolition waste | — | — | 77–110 | — | 110–190 | <5 As aggregate |
| Lime | 1000 | 200[h] | 777 ± 13 | — | 1165 ± 19 | — |
| Ultrabasic mine tailings | 8–250 | — | 40–250 | <50 | 60–377 | — |
| Hard coal ash | 20,000 | (2000–2600)[i] | 36 ± 6 | 20–30 | 73 ± 10 | |
| Lignite ash | | | 146 ± 28 | 230–264 | 246 ± 52 | |
| Marine algae biomass ash | | | 31 | — | 348 | |
| Wood/woody biomass ash | | | −89–815 | | −118 to 1766 | ~100. Up to 700 in high substitution specialised cements |
| Herbaceous and agricultural biomass ash | 490 | <−16,200 | −239–520 | 80–380 | −323 to 1505 | |
| Animal biomass ash[38] | | | 56–376 | — | 145–724 | |
| Biomass average | | | 186 ± 126 | — | 461 ± 260 | |
| Red mud | 5400 | (1080) | 47 ± 8 | 7–53 | 128 ± 18 < 440 with acid neutralising capacity of liquor | — |

Input data are presented in Supplementary Table 2 and Supplementary Note 1, all units in kg $CO_2$ $t^{-1}$
[a]Calculated by dividing the emissions of the production process by the mass of alkaline material
[b]Predicted future emission normalised to mass of alkaline material
[c]Maximum $CO_2$ capture potential by forming carbonate minerals
[d]$CO_2$ capture measured in experimental work
[e]Maximum enhanced weathering $CO_2$ capture potential
[f]$CO_2$ mitigation potential from other uses of material
[g]See Supplementary Notes 2 and 3
[h]Based on an 80% emission reduction target[26] (e.g., UK and EU)
[i]Accounting for aggregate primary energy carbon intensities in RCP2.6 by 2050. Brackets denote 2100 projected

Residue from coal and biomass combustion (e.g., fly and bottom ash) has been shown to carbonate in HTP experiments[27]. Due to the small particle size, large surface area and the high concentration of silica, ash is readily reused as a pozzolan or binder substitution in cement production, resulting in a saving of 100–700 kg $CO_2$ $t^{-1}$ over raw material[28], although the extent of substitution is limited by impact on strength. Furthermore, biomass ash has a long history of being spread onto agricultural land as an alternative liming agent[29]. Under the representative concentration pathway 2.6 (RCP2.6, the pathway most likely to result in <2 °C of warming), the emissions intensity of primary energy is predicted to decrease to 25 kg $CO_2$ $GJ^{-1}$ by 2050 and −11 kg $CO_2$ $GJ^{-1}$ by 2100[30]; the lower negative value is a result of biomass energy carbon capture and storage. As such, carbonation or enhanced weathering of ash (up to 1800 kg $CO_2$ $t^{-1}$) from biomass power generation could represent a non-trivial additional carbon draw-down (Supplementary Note 5 and Supplementary Table 2). The elevated phosphorus or sulphur content could limit the carbonation of some biomass ashes, resulting in an emission of $CO_2$ through the release of acidic waters during weathering.

Aluminium is produced by digesting bauxite ore in a high temperature solution of sodium hydroxide (known as the Bayer process), the products of which are alumina and a waste residue described as red mud. Red mud is composed primarily of iron and aluminium oxide/hydroxide, carbonate or hydroxide calcium (tricalcium aluminate, hydro-calumite) or sodium (sodalite, cancrinite) aluminates[31]. The residue is typically deposited with unreacted sodium hydroxide solution or dry-stacked. Carbonation of red mud has been demonstrated in HTP and ambient reactions, although only minor uptake was measured (<50 kg $CO_2$ $t^{-1}$)[32,33]. The maximum capacity of unsintered/causticised

red mud is 128 kg $CO_2$ $t^{-1}$. The supply of divalent cations through the addition of gypsum, calcium chloride or lime (e.g., during sintering) may further increase the carbonation of the residual solution NaOH and Na-aluminate minerals[34] (Supplementary Note 6).

Carbon uptake has been demonstrated in the waste materials and tailing ponds from asbestos[16], nickel[35] and diamond[36] mines, and in HTP experiments[37]. To estimate the carbon sequestration potential, we have focussed on the waste rock production from nickel and platinum group metal (PGM) mining. We have not accounted for waste from asbestos production, the future generation of which may be limited (Supplementary Note 7).

**Alkaline material production forecast**. By combining material economic saturation trends (see, e.g., van Ruijven et al.[38]) with forecasts of global economic development, consumption, population and biomass/coal primary energy from shared socioeconomic pathways (SSPs) and associated aggregated RCPs[39], it is possible to estimate future production of alkaline materials. We focus specifically on the contemporary and future production of alkaline materials, but there may also be tens of Gt of material stockpiled from historical production[40,41]. Figure 1 shows annual per capita production/consumption of cement, steel, PGM and nickel, and lime as a function of gross domestic product (GDP) or gross world product (GWP). Nonlinear least squares regression through national and global data were used to predict future production (a list of nations is included in the Supplementary Note 8). Consumption in the SSPs has been normalised to 2005 and used to derive relative changes to the baseline.

Maximum (SSP2) 2100 production estimates for cement, steel, aluminium and lime are 7.5 ± 0.4 Gt $yr^{-1}$, 7.1 ± 0.1 Gt $yr^{-1}$,

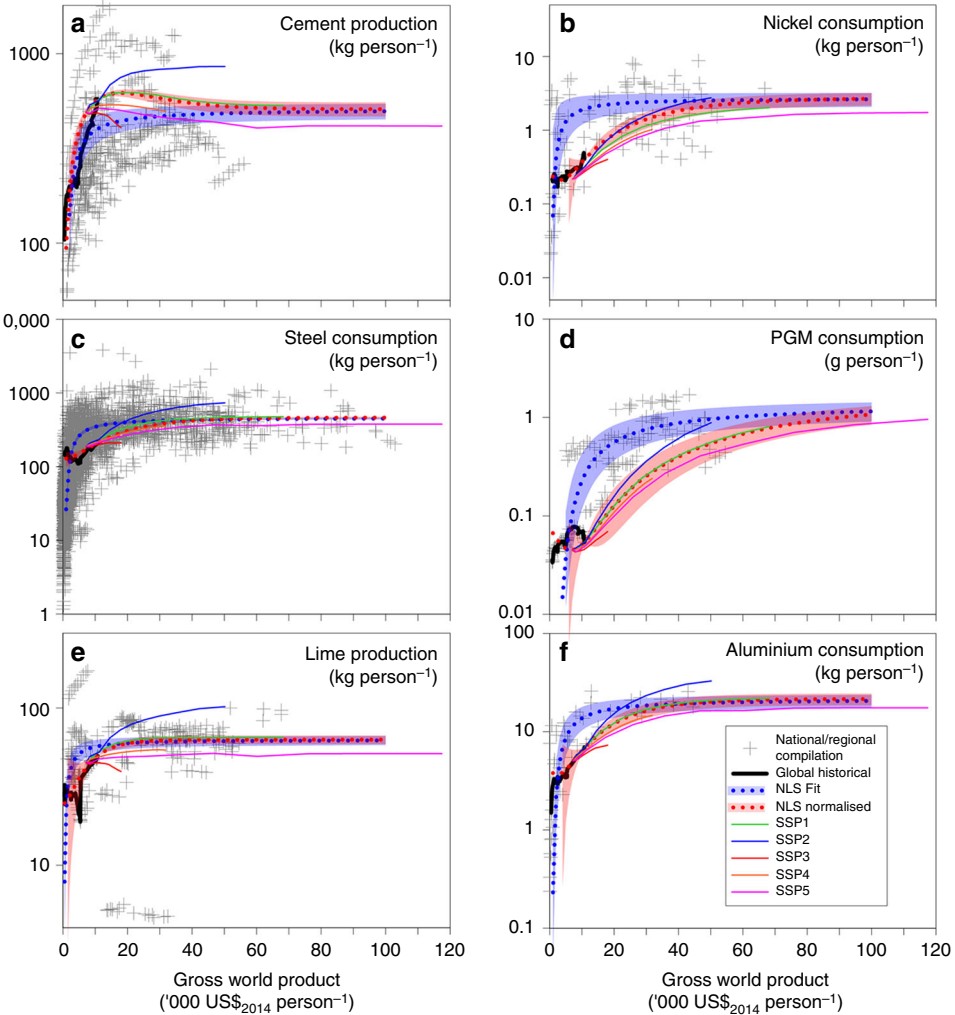

**Fig. 1** Consumption/production global saturation estimates for alkaline materials. **a** Cement, **b** nickel, **c** steel, **d** platinum group metal (PGM), **e** lime and **f** aluminium as a function of gross world product (GWP). The diagrams show a nonlinear least squares regression through compiled national data (blue dotted, the shading represents ±the standard error). The saturation value from this was fixed in an additional regression using global data relative to 2014 consumption (red dotted). Using the global fit as a baseline, the relative consumption projections for the shared socio-economic pathways (SSPs) were derived by normalising absolute changes in consumption. Production has been used for lime and cement that have a relatively small international trade market (<5%), otherwise apparent consumption has been plotted using national (slag) or regional (aluminium, PGM, nickel) data

$334 \pm 34$ Mt yr$^{-1}$ and $900 \pm 35$ Mt yr$^{-1}$ respectively (Fig. 2). Approximately 8–15% of the cement production is kiln dust which equates to between 245 Mt yr$^{-1}$ and 1.1 Gt yr$^{-1}$ by 2100. Approximately 300 Mt of concrete demolition waste are currently produced annually from a concrete stock of around 315 Gt[42]. Our model predicts production of demolition waste may increase to 20–40 Gt yr$^{-1}$ by 2100. Steel and blast furnace slag production may increase to 2.2 and 0.7 Gt yr$^{-1}$ respectively by the end of the century. Red mud from aluminium production may increase from 150 Mt yr$^{-1}$ currently to 500–1100 Mt yr$^{-1}$ by 2100. Primary energy from coal combustion in the SSP baseline scenarios is anticipated to vary between recent production 120 EJ yr$^{-1}$ to >880 EJ yr$^{-1}$. The RCP compilations largely predict decreases in coal use to <60 EJ yr$^{-1}$ for 2.6. Assuming a coal mix that changes from current levels (10% lignite, and 90% hard coal of bituminous/anthracite, with ash contents ~10%) to zero lignite by 2100, the total ash production varies between 130 Mt yr$^{-1}$ and 4.2 Gt yr$^{-1}$. An inverse relationship is predicted for biomass energy production, with ash production ranging from 300 Mt yr$^{-1}$ (SSP5) to 1.2 Gt yr$^{-1}$ in the RCP2.6 compilation. Up to 3.5 Gt yr$^{-1}$ of ultrabasic mine tailings (SSP2) may be produced by 2100 because of extracting ~25 Mt yr$^{-1}$ of nickel and ~5 kt yr$^{-1}$ of platinum group elements (see Supplementary Figs. 1–5 for material-specific production forecasts and associated carbonation potential).

## Discussion

The results suggest that the global $CO_2$ carbonation potential may increase from 1 GtCO$_2$ yr$^{-1}$, which is consistent with previous estimates based on current production[40], to between 2.3 and 3.3 GtCO$_2$ y$^{-1}$ in 2050 and 2.9 and 5.9 GtCO$_2$ yr$^{-1}$ by 2100 (Fig. 3). Trends in material consumption (high in SSP2, status in SSP5) drive the larger difference between these scenarios, with relative changes in GDP or population between other scenarios diminishing the difference in $CO_2$ capture potentials. Global $CO_2$ emissions in the baseline SSP scenarios in 2100 range from 24 GtCO$_2$ yr$^{-1}$ (SSP1) to 126 GtCO$_2$ yr$^{-1}$ (SSP5). Carbonating alkaline waste materials may mitigate between 5 and 12% of these baseline emissions. The lower emission RCPs predict $CO_2$ emissions to reach zero later this century and become net negative by up to 16 GtCO$_2$ yr$^{-1}$ (RCP2.6) in 2100[3]. As such,

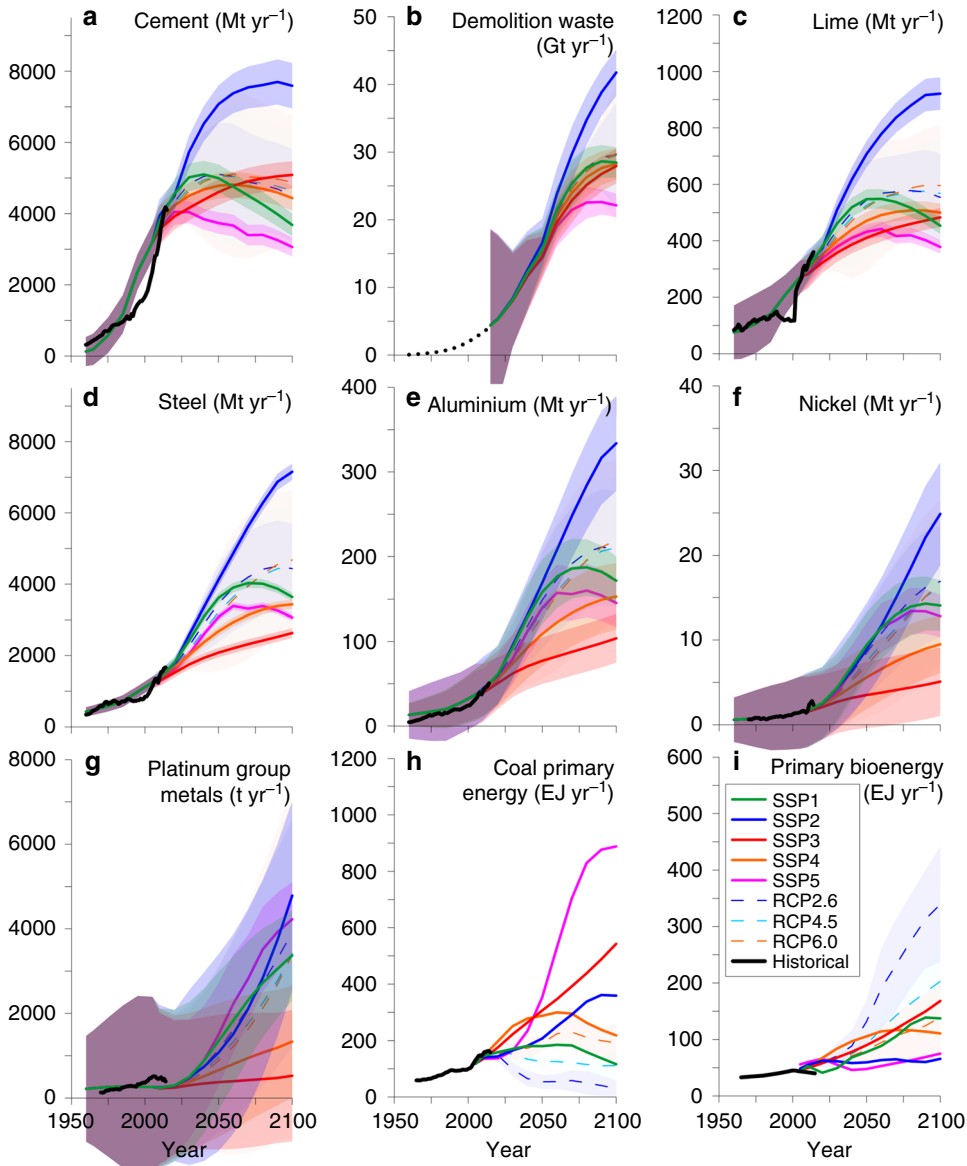

**Fig. 2** Production estimates for alkaline materials. **a** Cement, **b** demolition waste, **c** lime, **d** steel, **e** aluminium, **f** nickel, **g** platinum group metals, **h** coal primary energy and **i** primary bioenergy. Historical material production[58] and energy use[49] are also shown. Production forecasts were generated by combining a gross world product-per capita production saturation model, with projections of future economic growth, relative consumption, population and energy production

the carbonation of alkaline materials using atmospheric $CO_2$ could contribute ~18 and 37% of the negative emission requirements in RCP2.6. The enhanced weathering potential of alkaline materials (see Supplementary Fig. 6) ranges between 2.6 and 3.8 $GtCO_2$ $yr^{-1}$ in 2050, and increases to between 4.3 (SSP5) and 8.5 (SSP2) $GtCO_2$ $yr^{-1}$ by 2100. This is comparable to estimated potentials of other methods of removing $CO_2$ from the atmosphere. For instance, a recent synthesis report from the National Academy of Sciences, Engineering, and Medicine[2] suggest safe global scalable levels of sequestration to be 1–1.5 $GtCO_2$ $yr^{-1}$ for afforestation or forest management, 3 $GtCO_2$ $yr^{-1}$ for soil carbon management and 3.5–5.2 $GtCO_2$ $yr^{-1}$ for biomass energy carbon capture and storage. However, the land requirements of $CO_2$ capture using alkaline materials are considerably less.

These projections represent a theoretical maximum potential, which, in practice, would be difficult to realise. Research

investigating carbon uptake using these materials has primarily focused on HTP reactor experiments[11–13,27,32,37], which are typically far from optimised. Considerably more research is required to assess the potential for optimising $CO_2$ capture at ambient conditions.

Production data for many of these materials are typically not reported, and inventory assessments of current stockpiles are unlikely to be publicly available. As such, it is only possible to estimate production, as we have done, through proxy information. A more robust accounting mechanism is required to accurately assess the potential of alkaline materials.

Furthermore, there is no national or international mechanism for accounting for the value of $CO_2$ capture in waste. While this may be relatively trivial for carbonating materials emanating from a production site, it is more complicated for cement and lime where the latent carbon sequestration potential is only realised after many years of service life.

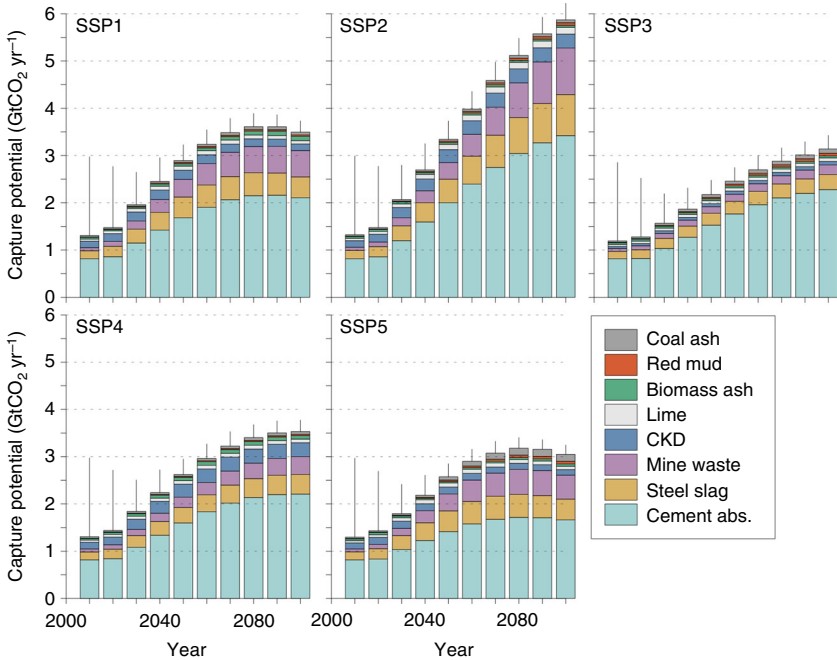

**Fig. 3** Forecast of $CO_2$ capture potential through carbonation of alkaline materials to 2100 for the baseline shared socio-economic pathways (SSPs). The error bars represent the standard error for the range of concentration pathways in the SSPs ($n = 4$ for SSPs 1 and 3, and $n = 5$ for SSPs 2, 4 and 5) together with uncertainties of material production and consumption, and chemistry

The economic cost of capturing $CO_2$ using alkaline materials could be relatively low as most are available as wastes or low-value by-products, and typically in particle sizes that facilitate rapid reaction. There may be additional processing costs (particularly in supplying $CO_2$ or water to the reaction site), which may lower the efficiency of the proposals. These costs should be explored further and included within integrative assessment models to consider the wider carbon balances of reacting atmospheric $CO_2$ alkaline wastes.

Before deployment at scale it is imperative that the environmental and social consequences of these activities are explored. Carbonate formation in alkaline waste materials has long been associated with lowering their environmental burden[22], whereas ash and slag have been used positively as replacement lime for agriculture. However, these materials are heterogeneous, and individual production sites will require unique and ongoing assessments.

Exploiting opportunities in existing industries for atmospheric $CO_2$ sequestration may contribute significantly to preventing climate change, by storing carbon permanently in either mineral carbonates or as dissolved bicarbonate in the ocean. It would be unwise to explore this potential at the expense of extensive emissions reduction. However, meeting the material demands of a growing global population will present an opportunity for low-cost atmospheric carbon dioxide sequestration that would be equally myopic to ignore.

## Methods

**Production forecast model.** A model that relates national or regional per capita material production (for cement and lime) or consumption (for aluminium, steel, platinum group metals, and nickel) (P) to per capita GDP (national or regional data, see Supplementary Fig. 7 and Note 2)[38] was regressed through historical data using nonlinear least squares (Eq. (4)).

$$P = ae^{-b/GDP}, \qquad (4)$$

where $a$ and $b$ are regression constants. The values returned for $a$ and $b$ for each material are included in the Supplementary Table 3 with their standard error. The derived saturation value, $a$, was used in a further regression through global data

normalised to 2014 production and GDP (Eq. (5)).

$$P = P_{REF} \times \left(1 + ((m + r) \times \Delta GWP) \times e^{\left(a \times \left(1 - e^{-(m \times \Delta GWP)}\right)\right)}\right) - (m \times \Delta GWP), \qquad (5)$$

where $P_{REF}$ is the global per capita consumption at a given reference year (2014), $\Delta GWP$ is the deviation of the per capita GWP from the reference year; and $m$ and $r$ are regression constants, for which $m$ was fixed and $r$ varied (a sensitivity analysis for variations in $m$ was performed to minimise the standard error for $r$). This formed the baseline which was modified with relative normalised consumption intensities ($C_t/C_{2005}$) for each SSP and RCP derivative. The per capita consumption model was combined with GWP capita$^{-1}$ and population forecasts (Pop) associated with the SSPs, to derive production forecasts (T) for cement, steel, aluminium, lime, PGM and nickel (Eqs. (6) and (7)).

$$P_{norm} = P \times \frac{C_t}{C_{2005}}. \qquad (6)$$

$$T(t) = P_{norm} \times Pop(t). \qquad (7)$$

Beyond the forecast change in economic consumption, we have not considered the penetration of recycling into metals production. Recycling would reduce the production of slag and remove completely the production of red mud and mine tailings. However, the proportion of material that may be recycled is limited (e.g., 69% for steel and 65% for aluminium[43,44], particularly for developing economies yet to reach saturation. As such, we may overestimate the contribution of $CO_2$ removal using slag, mine tailings or red mud. Cement, cement kiln dust, lime and ash have no capacity to be recycled as the original materials.

For every tonne of clinker, 115 ± 17 kg of cement kiln dust is produced as a by-product in kilns. While the latent capacity of $CO_2$ uptake in cement could be partially realised through elevated $CO_2$ curing (see, e.g., ref. [23]), we have only modelled carbonation/weathering through reabsorption during a 50-year service life (based on the method in ref. [24] and carbonation/weathering following demolition; see refs. [11,14], Supplementary Note 3, Supplementary Tables 4 and 5). Of the lime production, 20% was assumed available for reaction with $CO_2$ (see Supplementary Note 4). The ratio of pig iron to steel production (0.724 ± 0.002) was found using linear regression of 1960–2014 data, negating the need to explicitly model pig iron displacement from scrap recycling, assuming the scrap ratio remains unchanged. All steel and blast furnace slags were considered available for reaction with $CO_2$. While a substantial proportion of blast furnace slag is recycled as aggregate or for clinker replacement[18], we assume that the value (cost and carbon) is greater for reaction with atmospheric $CO_2$ than for clinker replacement (e.g., Table 1). If the silicon was extracted from the slag prior to carbonation, recycling and $CO_2$ capture may not be mutually exclusive. Between 2006 and 2014, there was 185 ± 5 kg of blast furnace slag and 117 ± 5 kg of steel slag produced for every tonne of crude steel[43]. Between 1967 and 2014, 3.5 ± 0.04 tonnes of red mud were produced for every tonne of aluminium (see Supplementary Note 6)[45]. Approximately 60% of nickel reserves are contained within ultrabasic laterite

deposits (containing $1.2 \pm 0.4\%$ s.d. Ni[46,47]), the remaining proportion is associated with nickel sulphide deposits (containing $0.4 \pm 0.4\%$ s.d. Ni[47]), the ratio of which we have assumed for future production. Approximately $81 \pm 24$ and $234 \pm 253$ tonnes of ultrabasic gangue (the non-commercial proportion of the ore) are produced for every tonne of nickel from laterite and sulfidic deposits respectively (Supplementary Note 7)[47]. Approximately 84% of base reserve PGM deposits are contained in ultramafic rock (containing $4.7 \pm 0.7$ g Pt and Pd t$^{-1}$ [48]), which has been used in this model. The remaining reserves are contained in nickel sulphide deposits and have not been considered to avoid double counting with the above. For every kg of PGM produced, $212 \pm 31$ tonnes of ore are processed.

Projections of future biomass and coal primary energy generation were taken from refs. [49], and combined with average higher heating values and ash contents (see Supplementary Note 5) to estimate future production of ash (Eq. (8)).

$$T(t) = E(t) \times HHV \times A, \quad (8)$$

where $E$ is the primary energy generation, HHV is the higher heating value and $A$ is ash content. For all materials, the production forecast is multiplied by the carbonation or enhanced weathering potential (Table 1).

**The $CO_2$ sequestration capacity of alkaline materials**. The carbonation ($C_{pot}$, Eqs. (1 + 3) and Eqs. (2 + 3), expressed in kg $CO_2$ t$^{-1}$) or enhanced weathering potential ($E_{pot}$, Eqs. (1) and (2)) for each material (example minerals are presented in Supplementary Table 6) was derived using the bulk elemental composition of iron and steel slag[50], cement, cement kiln dust[13], demolition waste[40], lime[51], coal ash[52,53], biomass ash[54], red mud[31] and PGM[55] and Ni[56] tailings in the modified Steinour formula[57] (Eqs. (9) and (10)).

$$C_{pot} = \frac{M_{CO_2}}{100} \cdot \left( \alpha \frac{CaO}{M_{CaO}} + \beta \frac{MgO}{M_{MgO}} + \gamma \frac{SO_3}{M_{SO_3}} + \delta \frac{P_2O_5}{M_{P_2O_5}} \right) \cdot 10^3, \quad (9)$$

$$E_{pot} = \frac{M_{CO_2}}{100} \cdot \left( \alpha \frac{CaO}{M_{CaO}} + \beta \frac{MgO}{M_{MgO}} + \varepsilon \frac{Na_2O}{M_{Na_2O}} + \theta \frac{K_2O}{M_{K_2O}} + \gamma \frac{SO_3}{M_{SO_3}} + \delta \frac{P_2O_5}{M_{P_2O_5}} \right) \cdot 10^3 \cdot \eta, \quad (10)$$

where CaO, MgO, $SO_3$, $P_2O_5$, $Na_2O$ and $K_2O$ are the elemental concentrations of Ca, Mg, S, P, Na and K, expressed as oxides (Supplementary Table 7), $M_x$ is the molecular mass of those oxides; coefficients $\alpha$, $\beta$, $\gamma$, $\delta$, $\varepsilon$, and $\theta$ consider the relative contribution of each oxide (Supplementary Figs 8 and 9); and $\eta$ is molar ratio of $CO_2$ to divalent cation sequestered during enhanced weathering. Equations (1) and (2) imply $\eta = 2$; however, due to buffering in the carbonate system, the value is between 1.4 and 1.7 for typical seawater chemistry, $pCO_2$ and temperature[8]. We have used $\eta = 1.5$, which is a conservative global average. The values of carbonation and enhanced weathering potential of alkaline materials is shown in Table 1, columns c and e, respectively.

Equations (9) and (10) imply that the potential is reduced by the presence of sulphur and phosphorus within the material. These elements are either bound to cations within the material, the dissolution reactions of which have no implicit reaction with $CO_2$ (Eqs. (11) and (12)), or they are present as, or may become, acid compounds which would impact the carbonate system to produce $CO_2$ (Eqs. (13) and (14)).

$$CaSO_4 \rightarrow Ca^{2+} + SO_4^{2-}. \quad (11)$$

$$Ca_5(PO_4)_3(OH) \rightarrow 5Ca^{2+} + 3PO_4^{3-} + OH^-. \quad (12)$$

$$H_2SO_4 + 2HCO_3^- \rightarrow SO_4^{2-} + 2CO_2 \uparrow + 2H_2O. \quad (13)$$

$$H_3PO_4 + 3HCO_3^- \rightarrow PO_4^{3-} + 3CO_2 \uparrow + 3H_2O. \quad (14)$$

## Data availability

Data generated as part of this study have been made available to download as supplementary data (Supplementary Data 1–9).

## Code availability

Coding data not applicable for this study.

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

## Acknowledgements

P.R. is funded by the UK's Greenhouse Gas Removal Programme, supported by the Natural Environment Research Council, the Engineering and Physical Sciences Research Council, the Economic & Social Research Council and the Department for Business, Energy & Industrial Strategy under grant no. NE/P019943/1.

## Author contributions

P.R. designed the study, undertook the calculations and wrote the manuscript.
