## [Peer Review File · Nature Communications]

Reviewers' comments:

Reviewer #1 (Remarks to the Author):

Synopsis

The correspondence "The negative emissions potential of alkaline materials" aims to quantify the amount of CO₂ that could potentially be removed each year from the atmosphere by using alkaline materials that are products or by-products of industrial process, coal and biomass combustion. The estimate the author presents is 2.5-7.5 GtCO₂ per year by 2100.

Recommendation

This correspondence will make an original contribution to the discussion around negative emissions, which encompasses both people from different research communities and people working on the design and implementation of decarbonization strategies with a view to stabilizing the global temperature increase at well below 2°C. It is to the best of my knowledge the first estimate of global negative emissions potentials of these pathways. I have some suggestions for improvements before proceeding to publication. Main comments and questions

- It would be excellent to put the estimated potential into perspective with other negative emission technologies and practices, which have recently been assessed in various outlets (e.g. National Academies of Sciences and Medicine 2018). Just to demonstrate that this really is an interesting alternative. (2050 potentials would also be nice!).
- The background has some flaws that can easily be amended:
 - o Lines 14 and 17: Integrated Assessment Models (IAMs) do not simulate different futures – the emission profiles coming out of IAMs are the result of cost-optimization.
 - o Line 15: Delete "overshoot and recovery", as this is also true for the low or no overshoot scenarios reaching 1.5°C (IPCC 2018).
 - o Line 16: Give a reference for the 750 Gt CO₂. In the recent IPCC report (see SPM), the range for 1.5°C is 100-1,000 Gt CO₂ cumulative (IPCC 2018).
 - o Line 22: What is meant by the phrase "poorly constrained feasibility"? That there are strong constraints? That they are less constrained? That the assessments of their feasibility are poor?
- The coverage of the different pathways is quite technical for a correspondence. It is hard to read for people without a background in chemical engineering.
- The description of Table 1 beginning in line 39 is not immediately intuitive. For example, why do we compare to column b for conventional mitigation potentials? Because it is 2050 and by then CO₂ intensity has improved due to deployment of renewables and CCS? It would also be helpful to give the units in Table 1.
- The results in Figure 3 are very interesting (by the way, why not show Gt CO₂ here?): Only in a business-as-usual world do we really see potentials in excess of 5 Gt CO₂. No matter whether a more sustainable pathway is then followed or whether we have fragmented, unequal or fossil-fuel 2 intensive developments, we do not get much beyond 3 Gt CO₂ (which is still substantial, of course). Is this really all because of higher material consumption in SSP2?
- The motivation of the correspondence is the reliance of IAMs on biomass (to produce energy combined with CCS) or afforestation to achieve negative emissions, which is both land-intensive.

However, the correspondence ends quite abruptly after presenting the estimates for the potentials, while there are a lot of other aspects that would need to be compared. An immediate question that jumps to mind is: Isn't this suggested method also much cheaper than BECCS, especially if based on by-products? And also the permanence should be much higher, especially compared to afforestation. And what about side effects? At least there should be an acknowledgement that these and other aspects should also be assessed in future research to close the loop with the beginning of the correspondence. (A way around squeezing some of this in without going over the word limit would be to add a table). Minor points

- Line 81: Substitute "for" by "from".
- Line 136: What do these numbers stand for? RCPs? But there is no RCP3.4.
- Line 153: Delete "appear".

References

IPCC 2018 Global warming of 1.5°C. An IPCC special report on the impacts of global warming of 1.5°C above pre-industrial levels and related global greenhouse gas emission pathways, in the context of strengthening the global response to the threat of climate change, ed V Masson-Delmotte, P Zhai, H O Pörtner, D Roberts, J Skea, P R Shukla, A Pirani, W Moufouma-Okia, C Péan, R Pidcock, S Connors, J B R Matthews, Y Chen, X Zhou, M I Gomis, E Lonnoy, T Maycock, M Tignor and T Waterfield National Academies of Sciences and Medicine E 2018 Negative Emissions Technologies and Reliable Sequestration: A Research Agenda (Washington, DC: The National Academies Press) Online: <https://www.nap.edu/catalog/25259/negative-emissions-technologies-and-reliablesequestration-a-research-agenda>

Reviewer #2 (Remarks to the Author):

The manuscript entitled "The negative emission potential of alkaline materials" is a highly topical work. It refers to the imminent global change of climate change and in particular in the feasible potential that the global production of alkaline materials has in decreasing the CO₂ concentration from the atmosphere. The author makes combined use of extensive databases with theoretical calculations. In addition, he discusses the actual decrease in CO₂ emissions that could be achieved based on the global production of alkaline materials from a variety of industries.

Probably this idea is not new, however in this article it is made an attempt to discuss the feasibility of such implementation in the real world based on actual data from the global production of these materials.

The suggested method is relatively straightforward. The potential benefit on global CO₂ emissions decrease is substantial with largely appreciable long-term consequences for the global ecosystem and short-term benefits for the public health. As someone that has worked with understanding the

CO2 capture capacity of similar materials I totally see the potential that the author is presenting in this work.

As mentioned already, the idea is not new. Here, in this work, it is made an attempt to use the platform that a prestigious journal such as Nature Communications offers to address this issue and use the opportunity for this idea to achieve higher visibility both in scientific but also non-scientific channels. With the hope, and expectation, that such a suggestion could reach policy-making actors, as Reviewer I support the idea of publishing such a work at a prestigious and highly credible journal. However such a potential comes with substantial responsibility towards the scientific community and the public. Therefore, as reviewer I will give a positive recommendation once the manuscript has been enriched with the suitable amount of references.

Futhermore, this is a work with strong potential to be used as a citation from a broad interdisciplinary areas. Therefore, I believe it is critical to provide adequate references to aide the cross-checking from the future reader, enhancing the usability of this manuscript.

In addition, after having a closer look in the manuscript, there are some issues that need to be suitably addressed:

Abstract

1) The first sentence is pretty long. It gives an excessively abstract feeling. I believe on line 3, the ", and" needs to be substituted by "as well as".

Main body

2) The passage on line 15 again does not make clear sense. Please amprove the phrasing.

3) For the reasons mentioned above, while reading I would appreciate a suitable reference at the end of the sentence on line 18.

4) The author mentions that the feasibility of CO2 removing technologies is poorly constrained. Is this a personal opinion? Is it an assessed fact? If yes, then it requires a suitable reference.

5) The sentence found between lines 24-27 is excessively long and rather confusing the way it is written now. The information provided in the parenthese is highly useful but the sentence gives the feeling that something is missing. It needs clear improvement.

6) The sentence in lines 27-30 needs to be written in a more clear way.

7) I believe that Table 1 needs to illustrate where these numbers are derived from. References seem to be missing.

- 8) The sentence found between lines 81-82 needs a suitable reference.
- 9) The acronym RCP2.6 seems not to have been defined earlier in the text.
- 10) Line 97, needs a reference.
- 11) Line 108, "SSPs" needs to be defined please.
- 12) Line 119: At which country does "national" refer to?
- 13) Lines 152-153: Again, SSP2 and SSP5 need to be defined.
- 14) The sentence on lines 161-163 requires a suitable reference.

Reviewer #3 (Remarks to the Author):

This manuscript evaluates the future potential for storage of carbon dioxide using industrially produced alkalinity and concludes that it could contribute significantly to achieving net negative CO₂ emissions by the end of this century. This is an area of considerable uncertainty and controversy (e.g., IPCC 2018). Thus the outcomes of this manuscript, which are supported by the data and analysis, are important and of general interest.

Renforth presents estimates of current rates of CO₂ storage through alkalinity and a projection of future rates based on several economic scenarios. In so doing, he lumps storage in aqueous and mineral phases. While there are individual estimates for current rates of CO₂ removal for these industries, I am not aware of projections to 2100. The results suggest that there is a significant opportunity for industrial alkalinity to contribute to negative emissions by the turn of the century. The value in this work is in the way that he has aggregated the various industries in a consistent way to allow for future projections. Thus this work will have impact in influencing how industrial alkalinity can be evaluated as our understanding of these systems improves. The result also validates further research efforts to clarify the scale and pathways for carbon storage.

He has not in any meaningful way addressed the fate and stability of CO₂ in that he does not differentiate between aqueous and mineral storage. In the process he is side-stepping important questions about other environmental and climate impacts. That said, I don't see a reasonable way for this to be addressed in this contribution.

A key factor in determining if industrially produced alkalinity will contribute to net negative emissions is the carbon cost of the industrial activities. Renforth presents results that would appear to accommodate as he reports contributions to net negative emissions, but it is not clear from the modeling how this is achieved. It appears that he is assuming that the upfront carbon costs are included in the emissions projections. How this is achieved and what energy sources and mixes contribute to this are not clear. This needs to be clarified.

Minor comments:

The treatment of ultra-basic mine waste involves some inconsistencies that should be corrected. I do not have the expertise to provide a similar evaluation of the other industrial streams.

Some comments relative to ms lines 218-227 that reflect a combination of confusion or omissions:

-It is not valid to count only laterite for Ni deposits as bedrock-hosted deposits are typically much more reactive. The sulfur content in them is typically many orders of magnitude smaller than the alkalinity, all of which can be accounted for in the model that Renforth employed.

- The source of some of the values used is not clear (e.g., line 223)

-On line 224 the author seems to equate all mine waste to waste rock, whereas mine waste consists of both waste rock and mine tailings (and other non-mineral waste streams).

The carbonation reactions presented in supplemental Information table S2 are not representative of the sources of alkalinity in mine waste. They should involve serpentine, pyroxene, talc, and olivine instead of brucite and periclase.

Table S3: this table omits the Mg in bedrock Ni mines which would typically be above 40 wt %, significantly larger than the values used for laterite and PGM ultra-basic rocks.

Finally, in Fig. S2 there appears to be an error labeling aqueous ionic Fe as Fe²⁺ rather than Fe³⁺.

Response to Review Comments

I would first like to thank the two anonymous reviewers and Prof. Dipple for spending time with the manuscript and offering suggestions for improvement. These suggestions were highly constructive and have helped to improve the manuscript for publication. I was delighted to see that all three reviewers had positive views on either the originality, scope, reach, methods, or conclusions.

Before going through the review comments in detail I can respond briefly to the highlighted comments from the Editor:

- strengthen the motivation by making more comparisons regarding other aspects other than the potential estimates (Reviewer #1).

The comparison to other negative emission technologies has now been made

- At the same time, we will need you to provide more analysis and discussion regarding the date and stability of CO₂ by differentiating between aqueous and mineral storage (Reviewer #3).

I have expanded the section that draws distinction between mineral carbonation and enhanced weathering. Essentially, this is distilling some key points from a review paper published on the subject.

- We will also need you to clarify the carbon costs of the industrial activities and energy sources and mixes (Reviewer #2 and #3).

This takes a little more explanation to respond to so please bear with me. The intention of this study is to present the carbon sequestration potential of alkaline materials based on a range of future socio-economic pathways (SSPs). It did not intend to set out the carbon emission pathways of the industries that produce them. For instance, Figure 3 presents the CO₂ draw-down for specific materials (e.g. slag), not the overall net balance of CO₂ for the industry (e.g. steel). The overall carbon balance of these industries will depend on a range of interconnected factors: technology assumptions/development, deployment of CCS, emissions in the power sector, penetration of low carbon fuels, and energy efficiency. The RCPs do not disaggregate individual industries, and construction of industry specific models is a considerable undertaking (and fraught with assumptions of those parameters).

However, not to completely side-step the issue, Table 1 shows that if industries pursued extensive emissions mitigation it may be possible for some of these industries to become net negative. The point being that the potential of alkaline materials is largely independent of the emissions scenario. If this potential was applied to the baseline (unmitigated) global emissions scenarios it may be able to counteract only 5-12% of these global emissions. However, if it were applied to an emission scenario that followed RCP2.6 (in which industrial emissions are already simulated), it could contribute between 18 – 37% of the global negative emissions requirements.

I've worked some of this explanation into the manuscript, together with improving the explanation of Table 1.

Reviewer 1 Comments	Response
It would be excellent to put the estimated potential into perspective with other negative emission technologies and practices, which have recently been assessed in various	Totally agree. Luck would have that the manuscript was submitted before the NAS, RS, or IPCC 1.5 report were available for

outlets (e.g. National Academies of Sciences and Medicine 2018). Just to demonstrate that this really is an interesting alternative. (2050 potentials would also be nice!).	citation. I've updated the introduction with these new reports. I've highlighted in the discussion the 2050 potentials, which can also be read from the diagrams. The potential presented here has also been compared to other methods of removing CO₂ from the atmosphere based on figures from the NAS report.
Lines 14 and 17: Integrated Assessment Models (IAMs) do not simulate different futures – the emission profiles coming out of IAMs are the result of cost-optimization.	Point taken, I've removed reference to simulation.
Line 15: Delete "overshoot and recovery", as this is also true for the low or no overshoot scenarios reaching 1.5°C (IPCC 2018).	Deleted
Line 16: Give a reference for the 750 Gt CO ₂ . In the recent IPCC report (see SPM), the range for 1.5°C is 100-1,000 Gt CO ₂ cumulative (IPCC 2018).	The original reference was Edenhofer et al., 2014, converting from C to CO ₂ . The range and reference to the IPCC 2018 report as you suggest is a better approach, so this has been used instead.
Line 22: What is meant by the phrase "poorly constrained feasibility"? That there are strong constraints? That they are less constrained? That the assessments of their feasibility are poor?	Language changed 'received substantially less attention compared with other proposals' with the reference moved to make it clear that it was one of the conclusions in Minx et al., 2018.
The coverage of the different pathways is quite technical for a correspondence. It is hard to read for people without a background in chemical engineering.	I've improved the explanation of the text in line with comments from the other 2 reviewers.
The description of Table 1 beginning in line 39 is not immediately intuitive. For example, why do we compare to column b for conventional mitigation potentials? Because it is 2050 and by then CO ₂ intensity has improved due to deployment of renewables and CCS? It would also be helpful to give the units in Table 1.	I've expanded the explanation of Table 1 for clarity and demonstration. More explanation of the figures in Table 1 has also been created in the Supplementary Information.
The results in Figure 3 are very interesting (by the way, why not show Gt CO ₂ here?): Only in a business-as-usual world do we really see potentials in excess of 5 Gt CO ₂ . No matter whether a more sustainable pathway is then followed or whether we have fragmented, unequal or fossil-fuel intensive developments, we do not get much beyond 3 Gt CO ₂ (which is still substantial, of course). Is this really all because of higher material consumption in SSP2?	Thanks, that is my interpretation also. The global potential of alkaline materials is a product of consumption, population, and economic development. Units changed to Gt CO₂ for consistency with the rest of the m/s
The motivation of the correspondence is the reliance of IAMs on biomass (to produce energy combined with CCS) or afforestation	I've expanded the discussion with these suggestions.

to achieve negative emissions, which is both land-intensive. However, the correspondence ends quite abruptly after presenting the estimates for the potentials, while there are a lot of other aspects that would need to be compared. An immediate question that jumps to mind is: Isn't this suggested method also much cheaper than BECCS, especially if based on by-products? And also the permanence should be much higher, especially compared to afforestation. And what about side effects? At least there should be an acknowledgement that these and other aspects should also be assessed in future research to close the loop with the beginning of the correspondence. (A way around squeezing some of this in without going over the word limit would be to add a table).	While I agree that this method of carbon sequestration is likely to be cheaper than other proposals, in the absence of engineering analysis, such statements are speculative. It would not be right to compare the cost too strictly between technologies at different TRL.
Line 81: Substitute "for" by "from".	Changed
Line 136: What do these numbers stand for? RCPs? But there is no RCP3.4.	Yes they stand for RCPs, text clarified. Good point, there was an intermediate forcing of 3.4 W/m ² used in the SSP modelling that has also been included in my analysis. I've removed reference to this as an RCP and from the figures.
Line 153: Delete "appear".	Deleted

Reviewer 2 Comments	Response
this is a work with strong potential to be used as a citation from a broad interdisciplinary areas. Therefore, I believe it is critical to provide adequate references to aide the cross-checking from the future reader, enhancing the usability of this manuscript.	Thanks! Good point regarding the references, which I've improved in line with Reviewer 1 first comment (broad reviews of NETs from RS, NAS and IPCC). I've also improved the references as you've pointed them out through the document.
The first sentence is pretty long. It gives an excessively abstract feeling. I believe on line 3, the ", and" needs to be substituted by "as well as".	Agreed. I've simplified the first sentence.
The passage on line 15 again does not make clear sense. Please amprove the phrasing.	I've removed the offending sentence, as suggested by Review 1.
For the reasons mentioned above, while reading I would appreciate a suitable reference at the end of the sentence on line 18.	A reference has now been included
The author mentions that the feasibility of CO2 removing technologies is poorly constrained. Is this a personal opinion? Is it an assessed fact? If yes, then it requires a suitable reference.	Language changed 'received substantially less attention compared with other proposals' with the reference moved to make it clear that it was one of the conclusions in Minx et al., 2018.

The sentence found between lines 24-27 is excessively long and rather confusing the way it is written now. The information provided in the parentheses is highly useful but the sentence gives the feeling that something is missing. It needs clear improvement.	Good point. The parentheses have been collected into their own sentence. This makes it a little easier to read.
The sentence in lines 27-30 needs to be written in a more clear way.	Thanks, I think I was too brief initially. I've expanded this description, and brought in implications for carbon storage as suggested by Reviewer 3.
I believe that Table 1 needs to illustrate where these numbers are derived from. References seem to be missing.	I've now referred to the supporting information, which now includes a section for the numbers in Table 1
The sentence found between lines 81-82 needs a suitable reference	This is in reference to Table 1, which has now been made clear.
The acronym RCP2.6 seems not to have been defined earlier in the text.	An explanation has been included here, and in the methods.
Line 97, needs a reference	Reference to supporting information has now been included.
Line 108, "SSPs" needs to be defined please.	Done
Line 119: At which country does "national" refer to?	A list has now been included in the supporting information
Lines 152-153: Again, SSP2 and SSP5 need to be defined.	These have now been expanded in the supporting information and a link to this included in the text.
The sentence on lines 161-163 requires a suitable reference.	This has been linked to a figure in the supporting information (Figure S8)

Reviewer 3 Comments	Response
Thus the outcomes of this manuscript, which are supported by the data and analysis, are important and of general interest.	Thanks.
He has not in any meaningful way addressed the fate and stability of CO2 in that he does not differentiate between aqueous and mineral storage. In the process he is side-stepping important questions about other environmental and climate impacts. That said, I don't see a reasonable way for this to be addressed in this contribution.	Thanks for pushing for this. I was too brief initially, as also highlighted by the other two reviewers. I've expanded the explanation of enhanced weathering and mineral carbonation. In a short piece like this, it is not possible to include much detail, but I've now added some of the key distinctions between the two mechanisms.
A key factor in determining if industrially produced alkalinity will contribute to net negative emissions is the carbon cost of the industrial activities. Renforth presents results that would appear to accommodate as he reports contributions to net negative emissions, but it is not clear from the modeling how this is achieved. It appears	I hope that my response above to the third comment from the Editor helps to explain the position of the paper. I've improved the explanation of Table 1, and highlighted the approach to carbon costs here.

that he is assuming that the upfront carbon costs are included in the emissions projections. How this is achieved and what energy sources and mixes contribute to this are not clear. This needs to be clarified.	
-It is not valid to count only laterite for Ni deposits as bedrock-hosted deposits are typically much more reactive. The sulfur content in them is typically many orders of magnitude smaller than the alkalinity, all of which can be accounted for in the model that Renforth employed.	Good point, this was based on an earlier decision before sulphur was included in the analysis. I've now included bedrock-hosted deposits, which was easy to do with the existing data, and updated the figures.
The source of some of the values used is not clear (e.g., line 223)	Reference is now included. Calculated from the ore grade, with Supplementary Note 8 explaining.
On line 224 the author seems to equate all mine waste to waste rock, whereas mine waste consists of both waste rock and mine tailings (and other non-mineral waste streams).	Good point, a misnomer on my part. What I was referring to was gangue of an ore, which ultimately ends up in tailings. I don't think it's possible to estimate the overburden from the Ni/ore grade proxy model I've used, this would depend more on the depth of the ore, which would be specific to each mine. I've cleared up the language in the text, in the online methods, and in the supporting information.
The carbonation reactions presented in supplemental Information table S2 are not representative of the sources of alkalinity in mine waste. They should involve serpentine, pyroxene, talc, and olivine instead of brucite and periclase.	Agreed. Olivine (as forsterite), and pyroxene (as diopside) have been included as examples of reactions in mine waste. I've retained brucite and periclase as potential reactions in some Mg rich slag. The list is meant to be illustrative rather than exhaustive.
Table S3: this table omits the Mg in bedrock Ni mines which would typically be above 40 wt %, significantly larger than the values used for laterite and PGM ultra-basic rocks.	This is now included along with the future projection of bedrock Ni mine tailings.
Finally, in Fig. S2 there appears to be an error labeling aqueous ionic Fe as Fe²⁺ rather than Fe³⁺.	Thanks for pointing this out. The figure has been corrected, and the supporting information expanded slightly for more explanation.

REVIEWERS' COMMENTS:

Reviewer #1 (Remarks to the Author):

Review Report

Recommendation

I thank the author for the comprehensive replies to my questions and comments and for implementing the suggestions I had made to improve the first version of the manuscript. Adding to the changes triggered by the other reviewers' comments, the manuscript has improved substantially: It is much more readable now for a wider audience of different backgrounds, the table is clear now making it much more useful, the introduction embeds the work much better in the current debate and state of knowledge, and the discussion (especially in the last paragraphs) is much more differentiated now, also pointing to the necessary research agenda. Therefore, I recommend publication. A couple of very minor issues (see below) can be fixed during editing.

Minor issues

Page 1, line 16: suggest to add after bracket "in 1.5°C pathways with little or no overshoot".

Page 1, line 16: Delete "in simulations" (because these are optimizations)

Page 3, line 56: IPPC = IPCC? 2018?

Page 11, line 188: ... become net negative....

Page 11, line 189: change "approximately" into "up to"?

Further remarks

Reviewer 3's remark about carbon costs of industrial activities and energy sources and mixes is understandable and well taken, as the first version of the manuscript was indeed not entirely clear on the use of the pathways and resulting potentials in the main text. What happens is that a purely technical/theoretical potential is estimated and the pathways/scenarios only serve as input for this estimation, mainly through the extent of material production. In my opinion, the explanation given by the author in the reply letter, but also the new representation of the approach in the manuscript clarify this much better now. If reviewer 3 is not satisfied, however, I could imagine that additionally pointing out the need to integrate the use of alkaline materials for withdrawal of CO₂ in an integrated assessment model (with industry) to find the ultimate carbon balances would further explicate what is done here (and more importantly what not).

Response to Review Comments

Review Comment	Response
Page 1, line 16: suggest to add after bracket "in 1.5°C pathways with little or no overshoot".	Changed to suit
Page 1, line 16: Delete "in simulations" (because these are optimizations)	Changed to suit
Page 3, line 56: IPPC = IPCC? 2018?	Yes, reference now included
Page 11, line 188: ... become net negative....	Changed to suit
Page 11, line 189: change "approximately" into "up to"?	Changed to suit
Pointing out the need to integrate the use of alkaline materials for withdrawal of CO ₂ in an integrated assessment model (with industry) to find the ultimate carbon balances would further explicate what is done here (and more importantly what not).	Changed to suite, Page 13 now reads: "The economic cost of capturing CO₂ using alkaline materials could be relatively low as most are available as wastes or low-value by-products, and typically in particle sizes that facilitate rapid reaction. There may be additional processing costs (particularly in supplying CO₂ or water to the reaction site), which may lower the efficiency of the proposals. These costs should be explored further and included within integrative assessment models to consider the wider carbon balances of reacting atmospheric CO₂ alkaline wastes."